# Electrochemical synthesis of peptide aldehydes via C–N bond cleavage of cyclic amines

Xinyue Fang[1,3], Yong Zeng[1,3], Yawen Huang[1], Zile Zhu[2], Shengsheng Lin[1], Wenyan Xu[1], Chengwei Zheng[1], Xinwei Hu[1] ✉, Youai Qiu [2] ✉ & Zhixiong Ruan [1] ✉

Peptide aldehydes are crucial biomolecules essential to various biological systems, driving a continuous demand for efficient synthesis methods. Herein, we develop a metal-free, facile, and biocompatible strategy for direct electrochemical synthesis of unnatural peptide aldehydes. This electro-oxidative approach enabled a step- and atom-economical ring-opening via C–N bond cleavage, allowing for homoproline-specific peptide diversification and expansion of substrate scope to include amides, esters, and cyclic amines of various sizes. The remarkable efficacy of the electro-synthetic protocol set the stage for the efficient modification and assembly of linear and macrocyclic peptides using a concise synthetic sequence with racemization-free conditions. Moreover, the combination of experiments and density functional theory (DFT) calculations indicates that different N-acyl groups play a decisive role in the reaction activity.

Peptide aldehydes are a significantly important class of biomolecules, playing a diverse range of essential roles in various living systems[1]. They serve as effective enzyme inhibitors, particularly protease inhibitors, such as Leupeptin[2,3], Elastatinal[4], and MG101[5,6] (Fig. 1A), thus regulating important biological processes such as digestion, blood clotting, inflammation as well as exhibiting anti-coronavirus activity[7]. Furthermore, aldehydes present in peptides provide a convenient handle for peptide backbone modification or site-specific ligation reactions[8–10]. However, due to the pronounced reactivity of aldehydes, late-stage incorporation in chemical synthesis is required through post-assembly chemical modification, either by employing a specific chemical reaction or by deprotection after peptide elongation to reveal the aldehyde[1,11–16]. Therefore, there is an ongoing and strong demand for innovative methods that enable highly efficient synthesis of peptide aldehydes.

Organic electrochemistry, utilizing protons and electrons as redox reagents, is emerging as a powerful approach for small-molecule synthesis[17–30]. However, it remains underexplored as a tool for achieving mild and chemoselective modification of existing peptides[31–33]. As a mild and customizable reaction platform[34], electrochemistry offers great promise in overcoming the chemo- and regioselectivity issues encountered in conventional bioconjugation strategies[35–39]. Recent contributions in electrochemical modification of peptides were mainly achieved by tyrosine-[40–45] and tryptophan-[46,47] specific functionalization as well as side chain diversification via direct or indirect electrochemical methods. Recognizing the necessity for additional research methodologies to facilitate the coupling of specific functional groups, various structural functionalization reactions of cyclic amines[48] were initiated, in order to access distinctive chemical and functional spaces. These reactions encompassed ring-opening reactions[49–51], ring-expansion reactions[52],

[1]Guangzhou Municipal and Guangdong Provincial Key Laboratory of Molecular Target & Clinical Pharmacology and the State Key Laboratory of Respiratory Disease, School of Pharmaceutical Sciences & the Fifth Affiliated Hospital, Guangzhou Medical University, Guangzhou 511436, PR China. [2]State Key Laboratory and Institute of Elemento-Organic Chemistry, Frontiers Science Center for New Organic Matter, College of Chemistry, Nankai University, 94 Weijin Road, Tianjin 300071, PR China. [3]These authors contributed equally: Xinyue Fang, Yong Zeng. ✉e-mail: xinweihu@gzhmu.edu.cn; qiuyouai@nankai.edu.cn; zruan@gzhmu.edu.cn

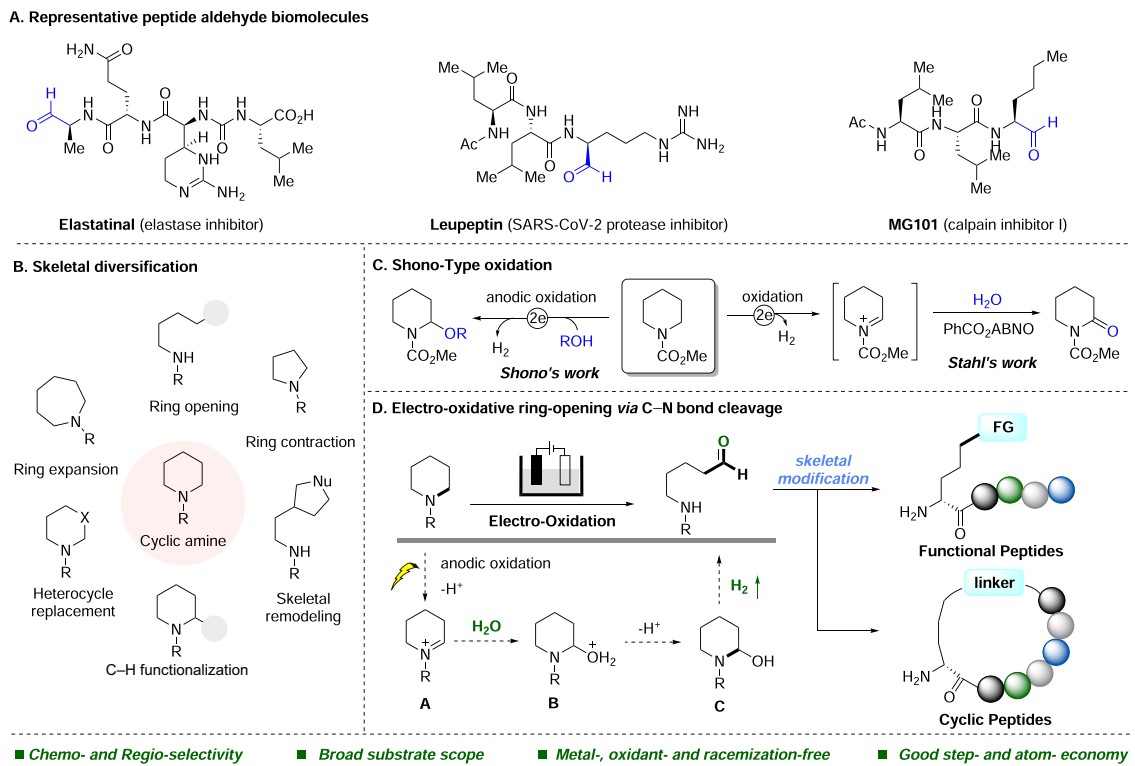

**Fig. 1 | Cyclic amine modification for peptide aldehyde synthesis. A** Representative peptide aldehyde biomolecules. **B** Skeletal diversification of cyclic amines (**C**) Shono-type oxidation. **D** This work: electro-oxidative ring-opening via C–N bond cleavage. ABNO = 9-Azabicyclo[3.3.1]nonane *N*-oxyl.

ring contraction[53], heterocycle replacement[54], and C2-direct functionalization reactions[55,56]. These innovative approaches involved the breaking of conventional saturated C–C or C–N bonds for chemical bond reorganization (Fig. 1B). In addition, with regard to piperidine derivatives, prevalent in pesticides and peptide-based pharmaceuticals[57,58], there is a wealth of examples employing the Shono-type oxidation strategy to achieve C2 functionalization of piperidines (Fig. 1C)[59–61]. However, there are few reports on electrochemical C–N bond ring-opening reactions. Within our program on sustainable electrochemical modification of peptides[62,63] and cyclic amines[64,65], herein we present an electro-oxidative ring-opening approach to achieve unnatural peptide aldehydes (Fig. 1D). Notable features of our general strategy include (i) (homo)proline-specific diversification of peptide backbones using exceedingly mild and biocompatible conditions in a metal-, chemical oxidant-, and racemization-free fashion, (ii) a broad substrate scope with various peptides and complex bioactive molecules, (iii) step- and atom-economical ring-opening procedure for smoothly yielding peptide aldehydes and remote amino aldehydes via selective C–N bond cleavage, and (iv) setting the stage for versatile syntheses of macrocyclic peptides. Detailed mechanistic insights of this process provided strong support for the presence and pivotal role of an *N*-Piv protecting group.

## Results

### Optimization of reaction conditions

We commenced our investigations by probing a variety of different electrolysis conditions, employing an undivided cell equipped with a graphite felt anode and a platinum cathode, for the envisioned electro-oxidative ring-opening of *D*-homoproline-containing dipeptide (Piv-Homopro-Leu-OMe) **1a**. The optimal results were obtained when dipeptide **1a** was directly electrolyzed at a constant current of 8.0 mA in a mixed electrolyte solution of $n$Bu$_4$NHSO$_4$ in MeCN/H$_2$O (9:1) at room temperature under atmospheric conditions without additional oxidants and transition metal catalysts. Under these conditions, the desired dipeptide aldehyde **1b** was isolated in 73% yield without

racemization (Table 1, entry 1) (For details, please see the Supplementary Table 1). The electrolyte and solvent were both found to be critical factors for the reaction to achieve the optimal yield (entries 2–6). Among a variety of electrolytes, $n$Bu$_4$NPF$_6$ and $n$Bu$_4$NOAc and $n$Bu$_4$NOH showed poor efficacy (entries 2–4), while LiClO$_4$, $n$Bu$_4$NI, or no electrolyte displayed inactive performance, highlighting the critical importance of HSO$_4^-$ (entries 5–6), which also confirmed by the further DFT calculations. In addition, solvent screening experiments verified the essential role of H$_2$O as the oxygen source, since performing the electrolysis in the solvent with traceless H$_2$O produced the remote amino aldehyde **1b** in extremely low yields (entries 7–9). The choice of electrode material proved critical because a dramatically reduced product yield was obtained when using other types of carbon electrodes (entries 10–12). A further control experiment indicated that electricity was indispensable to promote the desired reaction (entry 13).

### Substrate scope

With the optimal conditions in hand, we explored the scope of the electro-oxidative C–N bond cleavage protocol (Fig. 2). The electrochemical ring-opening reaction exhibited excellent compatibility with a variety of natural aliphatic amino acids such as leucine (Leu), alanine (Ala), valine (Val), threonine (Thr), glycine (Gly), and glutamic acid (Glu) as well as unnatural medicinally-useful bulky amino acids, e.g., L-*tert*-leucine, L-allylglycine, L-cyclohexylglycine, and cycloleucine, affording the dipeptide aldehydes in moderate to good yields (**1b**–**14b**). The structure of **11b** was unambiguously confirmed by single-crystal X-ray diffraction studies. Thereafter, we investigated the scope of tripeptides conjugated with a diversity of amino acid residues (**15b**–**22b**). The robust nature of the electro-oxidative ring-opening transformation was reflected by fully tolerating a wealth of valuable transformation groups, including alkenes and alkynes, which could serve as a handle for future late-stage bioconjugation. More importantly, the peptides containing electron-rich phenylalanine residues

**Table 1 | Optimization of reaction conditions[a]**

| Entry | Deviation from standard conditions | Yield [%][b] |
|---|---|---|
| **1** | **None** | **73** |
| 2 | $n$Bu$_4$NPF$_6$ instead of $n$Bu$_4$NHSO$_4$ | 43 |
| 3 | $n$Bu$_4$NOAc instead of $n$Bu$_4$NHSO$_4$ | 22 |
| 4 | $n$Bu$_4$NOH instead of $n$Bu$_4$NHSO$_4$ | 29 |
| 5 | LiClO$_4$ instead of $n$Bu$_4$NHSO$_4$ | 0 |
| 6 | $n$Bu$_4$NI instead of $n$Bu$_4$NHSO$_4$ | 0 |
| 7 | no $n$Bu$_4$NHSO$_4$ | 0 |
| 8 | MeCN/HFIP (9:1) instead of MeCN/H$_2$O (9:1) | 10 |
| 9 | MeCN instead of MeCN/H$_2$O (9:1) | 22 |
| 10 | C as an anode | 58 |
| 11 | RVC as an anode | 25 |
| 12 | GF as a cathode | 11 |
| 13 | No electricity | 0 |

Bold formatting shows that entry 1 is the optimal reaction conditions.

[a] Reaction conditions: Undivided cell, graphite felt (GF) anode, platinum plate (Pt) cathode, **1a** (0.3 mmol), $n$Bu$_4$NHSO$_4$ (0.3 mmol), MeCN/H$_2$O (9:1, 10 mL), constant current = 8.0 mA, 6 h (6.0 $F$), RT, under air. [b] Yields of isolated products. C = graphite plate. RVC = reticulated vitreous carbon. HFIP = 1,1,1,3,3,3-Hexafluoro-2-propanol.

were compatible under slightly modified electrolytic conditions, which normally readily decomposed under direct electrochemical conditions. To our delight, the strategy for the direct electrochemical synthesis of peptide aldehyde smoothly provided L-allylglycine-containing tetrapeptides and bulky alkyl group-containing pentapeptides, further emphasizing the strong biocompatible conditions (**23b–25b**). Furthermore, a wide range of common amide and ester substrates with bulky (*iso*-propyl, *tert*-butyl, cyclohexyl, adamantyl) or linear alkyl groups (*n*-butyl), including electron-donating (-OMe) and electron-withdrawing substituents (-CO$_2$Me, -CF$_3$, -CN), as well as synthetic useful handles (alkenyl, alkynyl, linker), were found to be fully tolerated by the optimized electrooxidation (**26b–44b**). Gratifyingly, the practical utility of our approach was further illustrated by successfully performing the desired ring-opening with substrates derived from marketed drug pregabalin, which effectively delivered $\varepsilon$-amino aldehyde derivative **35b** in moderate yield. It is noteworthy that the antioxidant activity of aldehyde products was evaluated by DPPH (2,2-Diphenyl-1-picrylhydrazyl) radical-scavenging activity assays, some peptides, such as **21b**, **22b**, and **24b**, exhibited mild antioxidant activity at a concentration-dependent manner (For details, please see the Supplementary Figs. 14 and 15).

Furthermore, encouraged by these exciting results, we sought to investigate the synthetic applications of the peptide aldehyde product by functional group transformation (Fig. 3). Thus, the outstanding potential of our ring-opening approach was demonstrated by its easy scalability and versatile transformations of the generated peptide aldehyde. For instance, we electrolyzed 5.0 mmol of dipeptide **1a** to deliver the corresponding peptide aldehyde **1b** in 72% yield, without appreciable loss in efficacy. In addition, five novel peptide sequences (**45–49**) were each rapidly constructed in simple steps from **1b**, highlighting the potential for such reactions to enable efficient diversification of native residues in preassembled peptide building blocks, including the synthesis of unnatural amino acids. Strikingly, a series of peptide-conjugated drugs (**50–56**) were further assembled through ligation with marketed drugs via amination or esterification.

In the realm of non-ribosomal peptide synthetase natural products, macrocyclic peptides stand out as highly prized therapeutic contenders compared to their linear counterparts. This preference is rooted in their heightened resistance to chemical and enzymatic degradation, improved receptor specificity, and advantageous pharmacokinetic profiles[49,66–69]. We sought to investigate the feasibility of efficiently constructing and expanding macrocycles by rapidly introducing new functional groups into peptides starting from a simple homoproline residue. Gratifyingly, olefin- and carboxylic acid-derived dipeptides (**48 and 49**) as well as tripeptide aldehyde (**22b**) could be rapidly transformed into four macrocycles containing (*E*)-alkene (**58**), amide (**59**), ester and alkyne (**60**)[70], as well as triazole (**61**) linkers, respectively, through concise synthetic sequences by key olefin metathesis, manganese-catalyzed C–H alkynylation, and click reaction steps (Fig. 4) (For details, please see the Supplementary Information on pages 69–80). The incorporation of functional groups into the linker region of stapled peptide-like structures has been demonstrated to influence the biological properties of the resulting products. Our post-assembly electro-oxidative strategy enables the rapid synthesis of a diverse array of functionally and structurally enriched molecules, as exemplified by the small library of compounds synthesized from dipeptide aldehyde **1b**.

Remote amino aldehydes, especially unnatural derivatives, serving as direct precursors of variant amino acids, are indispensable building blocks in the synthesis of peptide aldehydes[1,71,72]. Thus, the scope of cyclic amines[73–75] was also investigated using a slightly modified set of conditions (Fig. 5 and Supplementary Table 2 in Supplementary Information). Diverse substitution patterns on the piperidine ring exhibited excellent tolerance, enabling the synthesis of the corresponding acyclic amino aldehydes with moderate to good yields (up to 88%). Notably, benzoyl-protected piperidine afforded the corresponding aldehyde in moderate yield (**62b**), while that with other protected groups Boc and Cbz resulted in *ortho*-hydroxylation products instead of the desired aldehydes (For details, please see the Supplementary Table 3). Piperidines

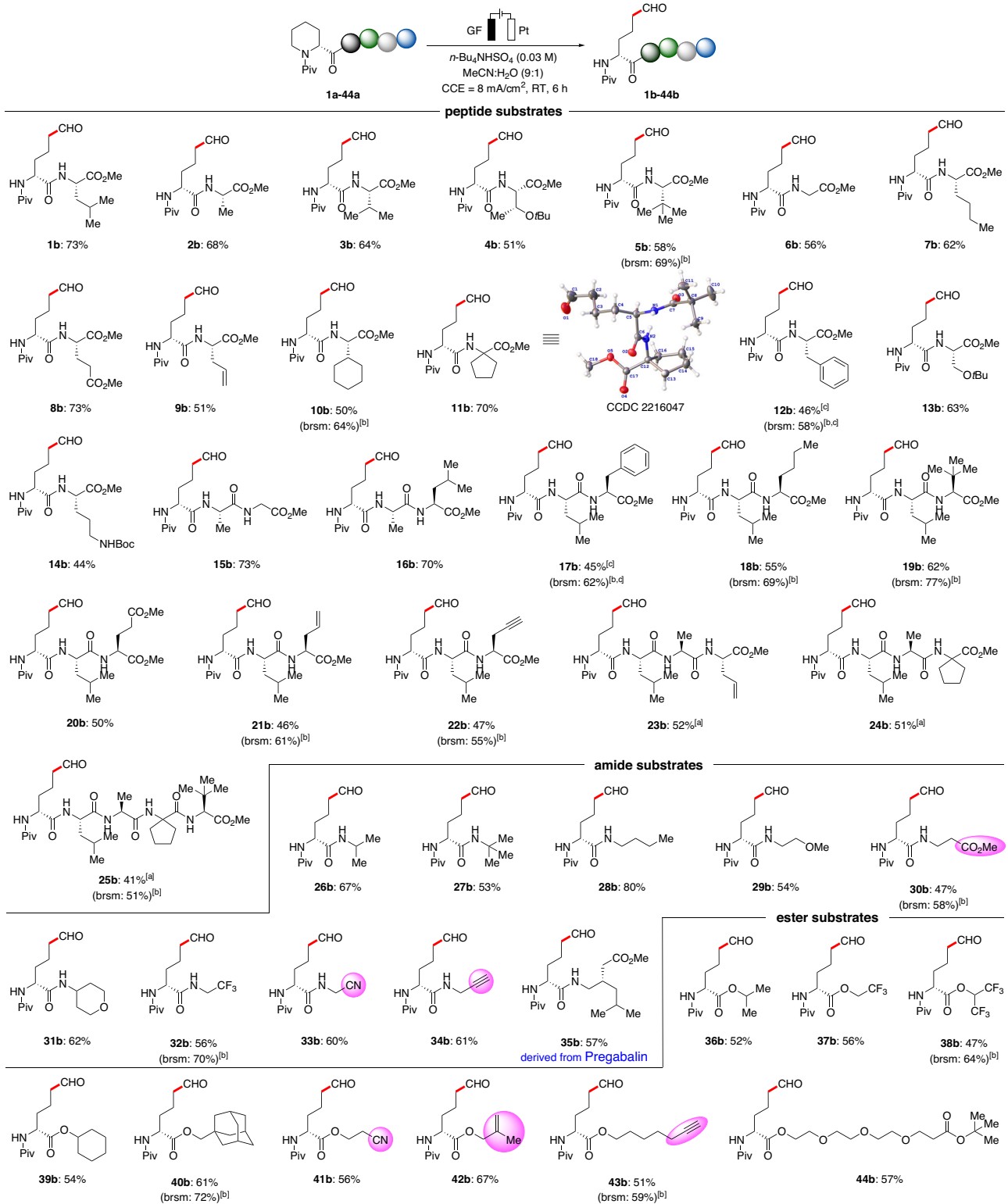

**Fig. 2 | Scope of electrochemical ring-opening with peptides, amides, and esters.** Reaction conditions: Undivided cell, graphite felt (GF) anode, platinum plate (Pt) cathode, **1a-44a** (0.3 mmol), nBu₄NHSO₄ (0.3 mmol), MeCN/H₂O (9:1, 10 mL), constant current = 8.0 mA, 6 h (6.0 F), RT, under air. Isolated yields are reported. [a] 9.0 h. [b] Yields based on recovered starting materials. [c] ketoABNO (50 mol %).

with ester functional groups on the saturated azacyclic backbone (**66a** and **68a**) exhibited effective performance. However, cyclic amines with free carboxylic acid groups at the α-positions (**69a** and **70a**) produced the respective decarboxylated amino aldehydes. Piperidines containing benzylic sites prone to oxidation yielded the

corresponding aldehydes with a 57% yield (**65b**). Remarkably, complete positional selectivity was observed with 2-substituted azacycles (**64a, 66a**, and **68a**) in the oxidative ring-opening protocol. The scalability of our ring-opening approach was demonstrated by a successful gram-scale reaction (**63b**). The method also

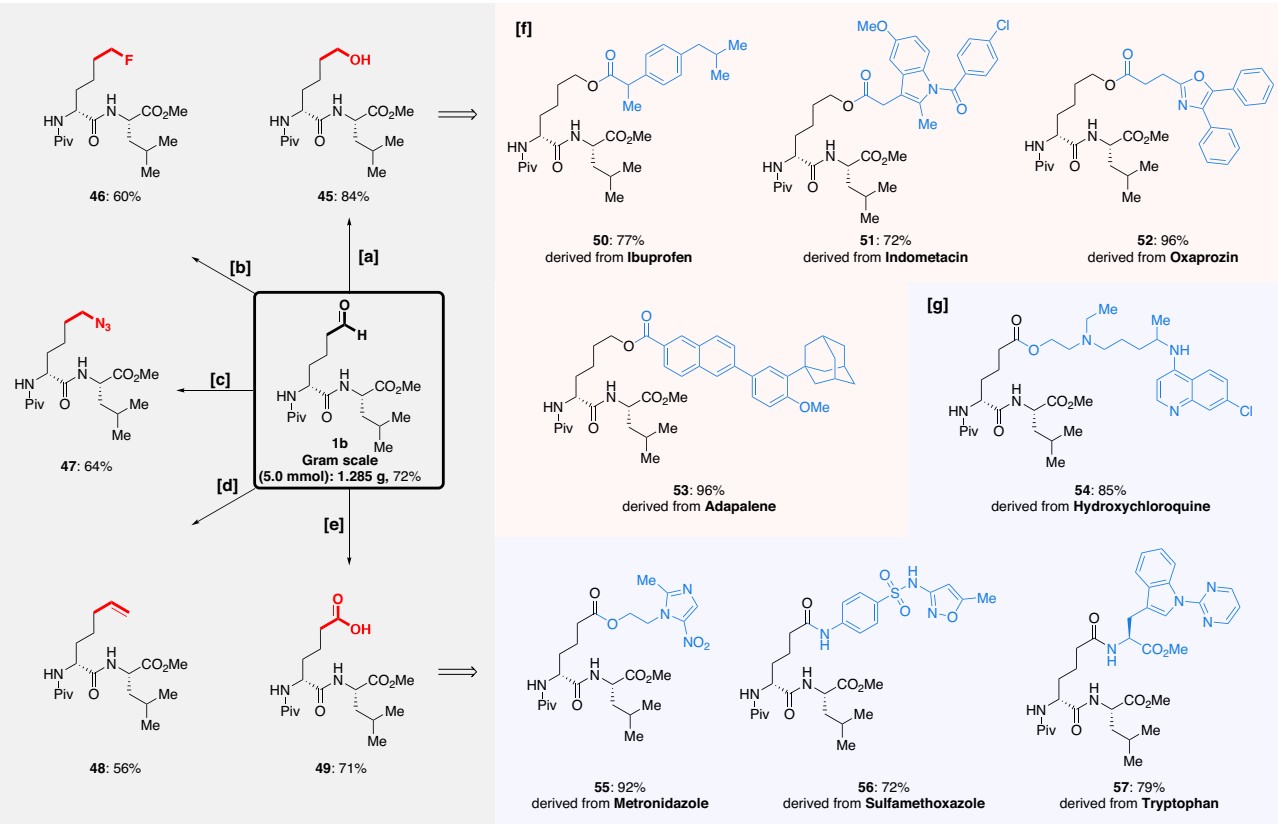

**Fig. 3 | Diversifications of desired peptide aldehyde.** Isolated yields are reported. [a] NaBH$_4$, MeOH, 0 °C, 1 h. [b] (i) NaBH$_4$, MeOH, 0 °C, 1 h; (ii) DAST, DCM, −78 °C to RT, 14 h. [c] (i) NaBH$_4$, MeOH, 0 °C, 1 h; (ii) MsCl, Et$_3$N, DCM, 0 °C to RT, 4 h; (iii)

NaN$_3$, DMF, RT, 12 h. [d] PPh$_3$MeBr, *t*-BuONa, THF, 0 °C to RT, 12 h. [e] O$_2$, NHPI, MeCN, RT, 12 h. [f] Transformations of hydroxyl peptides. [g] Applications of acid products. For details, please see the Supplementary Information on pages 57–69.

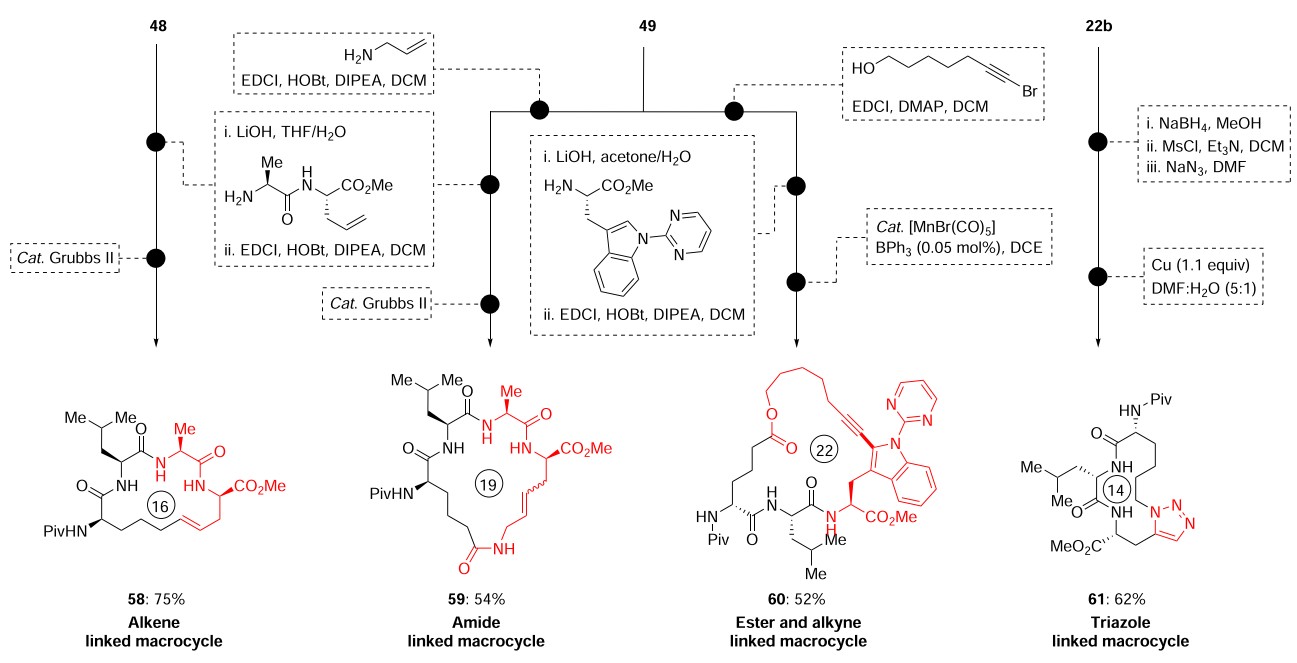

**Fig. 4 | The synthesis of macrocyclic peptides 58–61.**

tolerated saturated azacycles of various ring sizes (four to eight-membered rings), resulting in β-, γ-, or remote amino aldehydes (**67b**, **72b**, and **73b**), although the isolated yield for β-amino aldehyde (**71b**) was low. Moreover, the α,β-unsaturated aldehyde **74b**

was obtained when the piperidine's para-position was substituted with a methoxy group. Finally, both substituted cyclic amines **75a** and **76a** could be converted into corresponding ring-opening products, although the chemo-selectivity was unromantic.

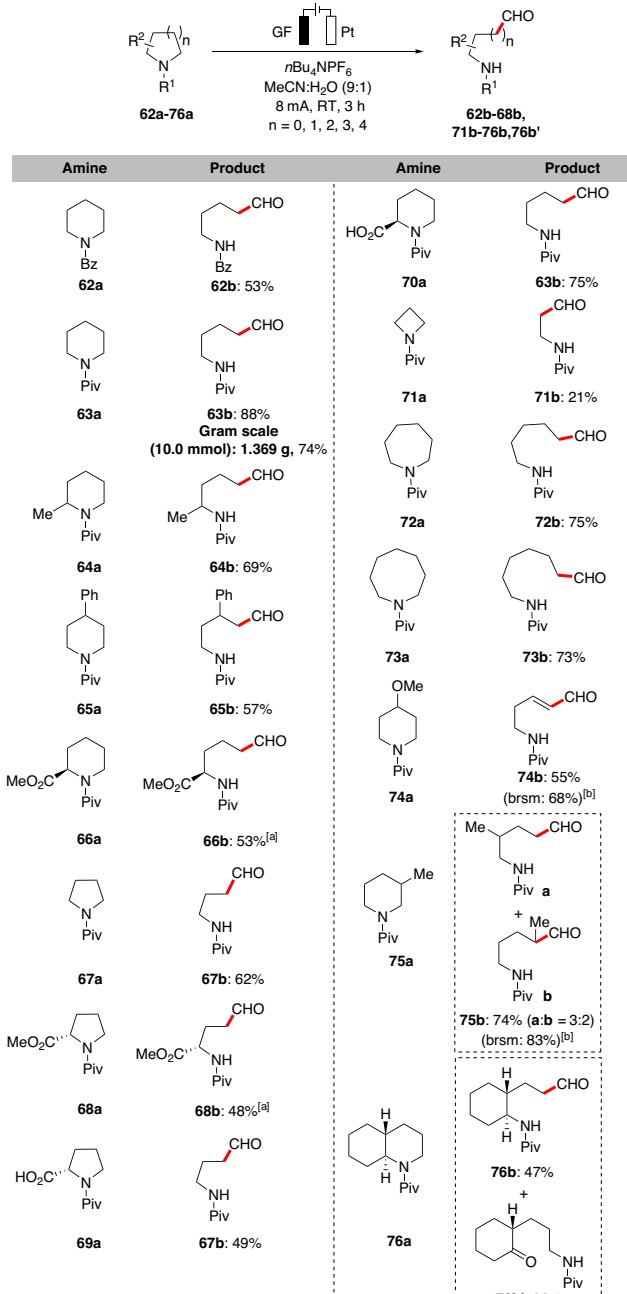

**Fig. 5 | Scope of simple cyclic amine derivatives.** Reaction conditions: Undivided cell, graphite felt (GF) anode, platinum plate (Pt) cathode, **62a-76a** (0.3 mmol), $n$Bu$_4$NPF$_6$ (0.3 mmol), MeCN/H$_2$O (9:1, 10 mL), constant current = 8.0 mA, 3 h (3.0 $F$), RT, under air. Isolated yields are reported. [a] 6.0 h. [b] Yield based on recovered starting materials.

## Mechanistic studies

In light of the outstanding versatility of the electrochemical ring-opening methodology, we were intrigued to delineate its mode of action. To this end, an isotopic labeling experiment was conducted in the presence of H$_2$$^{18}$O under the standard conditions. The results indicated that the oxygen atom in the newly formed aldehyde derived from the isotopically labeled water (Fig. 6A). Experiments with isotopically labeled co-solvents unraveled a facile C–H activation step, as evidenced by H/D scrambling at the $\alpha$-position of the aldehyde (Fig. 6B). Furthermore, analysis of the voltammogram results disclosed that the dipeptide **1a** underwent anodic oxidation at -1.928 V (vs Ag/AgCl), while no distinct oxidation peak was

observed for **1b**, indicating the difficulty of further oxidizing aldehyde **1b** to acid **49** (Fig. 6C). This observation provided a plausible explanation for the absence of carboxylic acid products in the electrolytic reaction, even when using higher currents. Additionally, an electricity on/off study revealed that any chain processes were transient and short-lived (Fig. 6D). This finding suggested that electrolysis is required for sustained product formation, indicating that the reaction was less likely to proceed through a radical chain mechanism.

Furthermore, DFT calculations were performed to further elucidate the reaction mechanism using **63a** as a model substrate (see Fig. 7 and Supplementary Table 7 in Supplementary Information). As is shown in Fig. 7, **63a** may first undergo two single electron anodic oxidation and the iminium intermediate **Int3** emerges. According to the potential energy surface, across **TS2**, water capture of **Int3** is endothermic ($\Delta G = 8.98$, $\Delta G^* = 14.70$). In the yielding **Int5**, an O−H···O intramolecular hydrogen bond is found to stabilize the intermediate, while the analogous intermediate from water capture of **Int3′** cannot be located. Taking HSO$_4$$^-$ in $n$Bu$_4$NHSO$_4$ as a Bronsted base, the resulting **Int5** may deprotonate barrierlessly into **Int6**, and reprotonate into **Int7** via **TS3**, dissipating 3.32 kcal/mol Gibbs free energy. The proton transferring is a fast process due to low energy barriers. A C−N bond cleavage will take place in **Int7**, mounting an energy barrier of 12.12 kcal/mol via **TS4**, yielding protonated product **Int8**, which may associate with HSO$_4$$^-$ into **Int9**. Moreover, another reaction pathway was also considered, in which the C−N bond cleavage takes place in **Int6** via **TS5** yielding an enol-like product. However, the energy barrier (26.72 kcal/mol) is too high to cross.

We then altered the *N*-acyl group into Boc, Bz, and Cbz, and calculated the relative Gibbs free energy of some corresponding intermediates and transition states. Results are listed in Table 2. In each variation, **Int3** is selected as the energy zero-point.

With DFT results in hands, the insight into reactivity of substrates can be gained. According to the proposed mechanism, applying the steady-state approximation and Eyring equation, a rate equation in proportion to the concentration of iminium (**Int3**) could be derived (To check the process of deduction, please see the Supplementary Information on pages 91 – 93):

$$r_{Int3\rightarrow Product} = k_{obs}[\text{H}_2\text{O}] \cdot c(\text{Int3}) \quad (1)$$

$$k_{obs} = \frac{k_B T}{h[1 + \exp(-\frac{\Delta G(Int3\rightarrow Int3')}{RT})]} \exp\left[-\frac{G(TS4) - G(Int3)}{RT}\right] \quad (2)$$

Where $r$ is the reaction rate of **Int3**'s conversion to the product form, $k_{obs}$ denotes the observed rate constant, $k_B$ is the Boltzmann constant, $T = 298.15$ K, $h$ is the Planck constant, $R$ is the gas constant, [H$_2$O] is the concentration of water, and $c$(Int3) is the analytic concentration of **Int3**, the iminium intermediate. According to the formula (Eq. 2), for each variation of the *N*-acyl group, the free energy difference between **TS4** and **Int3** plays a pivotal role in deciding the reactivity. Using the DFT-calculated data from Table 2, different *N*-acyl groups bear different $k_{obs}$ calculated in Table 3. Among Boc, Bz, Cbz, and Piv-substituted cyclic amines, the Piv-substituted reacts the fastest ($8.9 \times 10^{-2}$ L mol$^{-1}$ s$^{-1}$), but with an oxygen atom inserted (the Boc-substituted), $k_{obs}$ drops prominently to $7.2 \times 10^{-7}$ L mol$^{-1}$ s$^{-1}$, which implies the scarce reactivity.

## Discussion

In summary, we have reported a pioneering electrochemical ring-opening protocol that enables direct synthesis of unnatural peptide aldehydes and remote amino aldehydes through mild and biocompatible reaction conditions involving C–N bond cleavage. The strategy

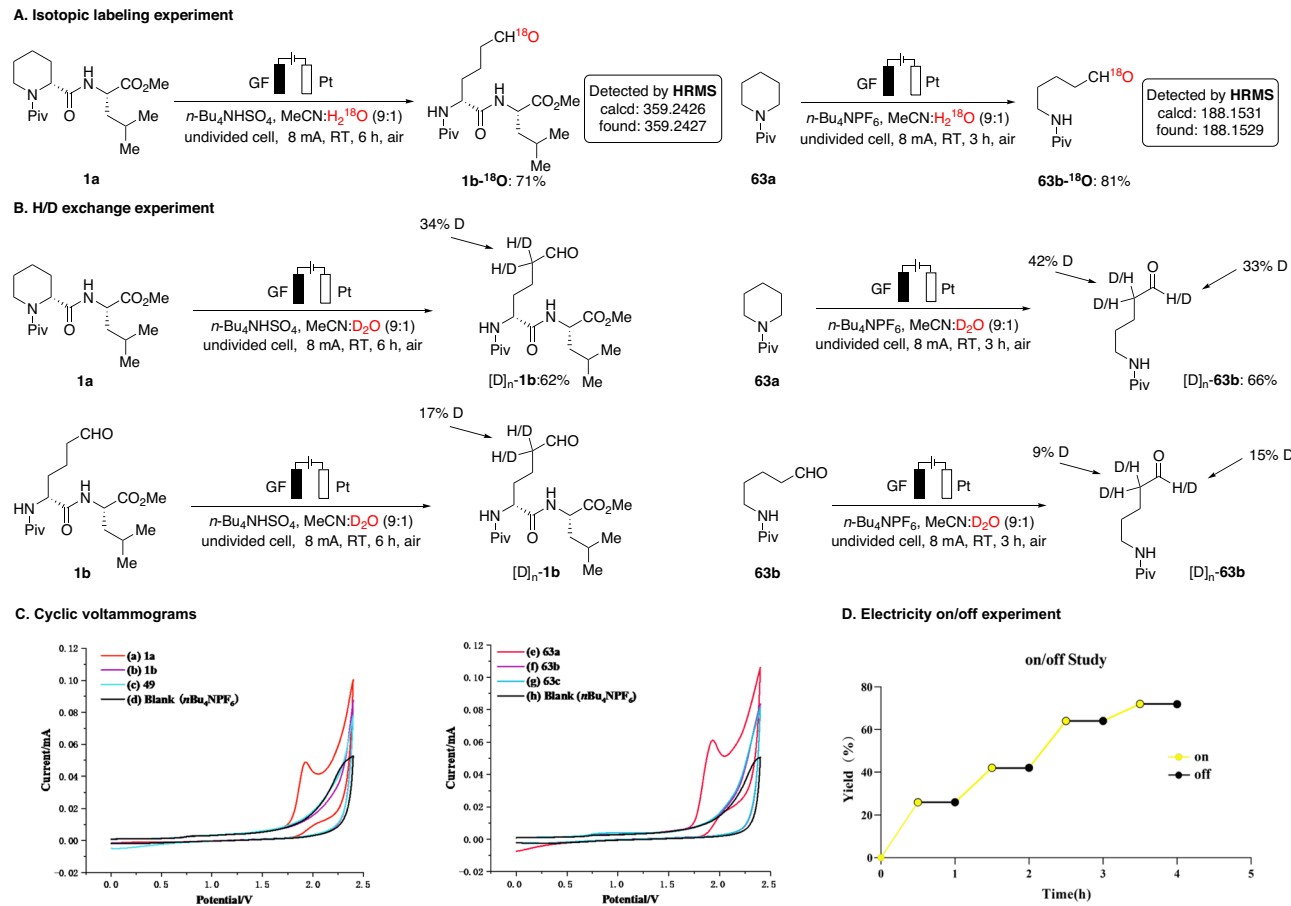

**Fig. 6 | Control experiments. A** Isotopic labeling experiment. **B** H/D exchange experiment. **C** Cyclic voltammograms. **D** Electricity on/off experiment.

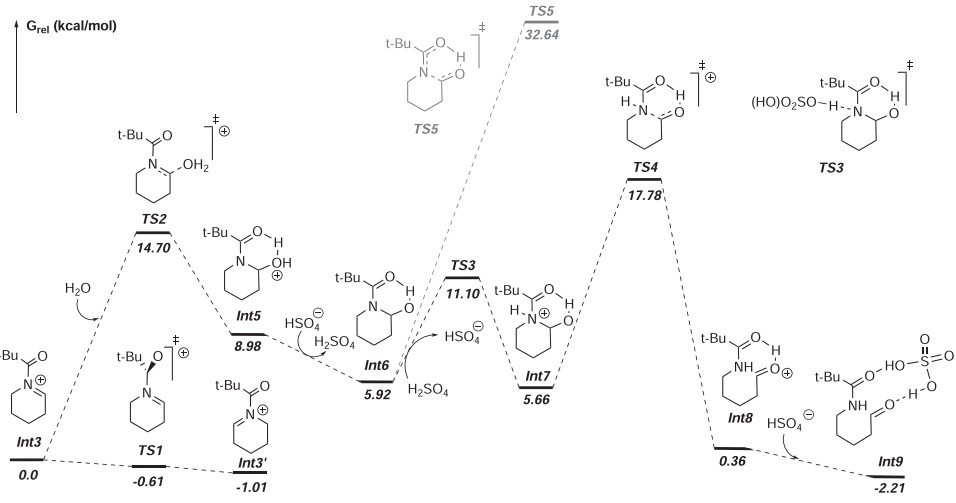

**Fig. 7 | DFT-calculated potential energy surface of Int3's conversion process.** All energy units are kcal/mol.

offers expedient access to a wide variety of peptide aldehydes with potential antioxidant activities, as well as expanded substrate scope including amides, esters, and cyclic amines with diverse synthetic functional groups. The unique power of the electro-oxidative ring-opening approach set the stage for efficient modification and assembly of acyclic and macrocyclic peptides by short synthetic sequence under racemization-free conditions. Notably, the experiments and DFT calculations demonstrate that different *N*-acyl groups play a decisive role for the reaction.

## Methods

### General procedure for electrochemical reactions

In an undivided cell (30 mL) equipped with a stirring bar, a mixture of substrates (0.3 mmol), $n$Bu$_4$NHSO$_4$ (0.3 mmol, 0.03 M), and MeCN/H$_2$O (9:1, 10 mL) were added. The cell was equipped with graphite felt plate (GF, 1.5 cm × 1.0 cm × 0.2 cm) as the anode and platinum plate (Pt, 1.5 cm × 1.0 cm × 0.01 cm) as the cathode connected to an AXIO-MET AX-3003P DC regulated power supply. The reaction mixture was stirred and electrolyzed at a constant current of 8 mA at room

**Table 2 | Relative calculated Gibbs free energy data of some corresponding intermediates and transition states, with the *N*-Piv group altered to Boc, Bz, and Cbz**

| *N*-acyl group & G$_{rel}$ (kcal/mol) | Int3′ | Int3 | TS2 | Int5 | Int6 | Int7 | TS4 |
|---|---|---|---|---|---|---|---|
| Boc | 0.22 | 0.0 | 16.10 | 13.24 | 6.58 | 10.77 | 23.63 |
| Bz | -0.82 | 0.0 | 14.31 | 9.47 | 5.10 | 8.08 | 20.67 |
| Cbz | 0.51 | 0.0 | 15.15 | 14.85 | 4.74 | 9.69 | 22.34 |
| Piv | -1.01 | 0.0 | 14.70 | 8.98 | 5.92 | 5.66 | **17.78** |

Bold formatting shows that the TS4 Gibbs free energy data for the *N*-Piv group are the lowest.

**Table 3 | Computed *k*$_{obs}$ of different Int3's conversion with varying *N*-acyl groups**

| Group | Boc | Bz | Cbz | Piv |
|---|---|---|---|---|
| *k*$_{obs}$/(L·mol$^{-1}$·s$^{-1}$) | $7.2 \times 10^{-7}$ | $8.8 \times 10^{-4}$ | $1.8 \times 10^{-4}$ | **$8.9 \times 10^{-2}$** |

Bold formatting shows that the Piv-substituted reacts the fastest.

temperature for 3-6 h. Upon completion, the solvent was removed directly under reduced pressure to afford the crude product, which was further purified by flash column chromatography to afford the desired products.

## Data availability

The authors declare that the data supporting the findings of this study are available within the paper and its Supplementary Information. Extra data are available from the corresponding author upon request. Source data are provided with this paper. Crystallographic data for the structures reported in this Article have been deposited at the Cambridge Crystallographic Data Centre, under deposition numbers CCDC 2216047 (11b). These data can be obtained free of charge from The Cambridge Crystallographic Data Centre via www.ccdc.cam.ac.uk/data_request/cif. Source data are provided with this paper.

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

## Acknowledgements

Support by the National Natural Science Foundation of China (22271067 to Z.R., 22201052 to X.H.), Key-Area Research Project of Guangdong

Provincial Department of Education (2022ZDZX2051 to Z.R.), Guangzhou Science and Technology Project (2023A04J0696 to X.H.), and the Plan on Enhancing Scientific Research in Guangzhou Medical University (GMU) (Z.R.) is most gratefully acknowledged.

## Author contributions

Z.R. directed the project and wrote the manuscript. Z.R., Y.Q., X.H., and X.F. conceived and designed the study and wrote the draft manuscript. X.F., Y.Z., Y.H., S.L., and C.Z. performed the experiments, mechanistic studies and analyzed the data. Y.Q. and Z.Z. performed the DFT calculations and analyzed the data. W.X. performed the bioactive assay. All authors contributed to scientific discussion.

## Competing interests

The authors declare no competing interests.
