## [Peer Review File · Nature Communications]

Electrochemical Synthesis of Peptide Aldehydes via C–N Bond Cleavage of Cyclic AminesREVIEWER COMMENTS

Reviewer #1 (Remarks to the Author):

In this manuscript, Hu, Qiu, Ruan, and their co-workers report an electrochemical strategy for ring-opening of cyclic amines via oxidative C–N bond cleavage. The major reaction scope includes simple piperidine rings containing a range of peptide motifs, as well as amides and ester appendages, which leads to the corresponding acyclic amino aldehydes under mild reaction conditions. The reaction is also effective for cyclic amine structures of differed ring sizes, such as azetidines, pyrrolidines, azepines, and azocanes. The synthetic value of this development has been proven by further modification of the furnished amino aldehydes via reduction, fluorination, azidation, olefination, oxidation, esterification, and synthesis of macrocyclic peptides. Furthermore, the authors carried out the mechanistic investigation using labeling experiments and DFT calculations.

Despite the extensive synthetic studies with a wide range of reaction scope and applications, the major concern arises on the originality of the presented chemistry. Oxidative C–N bond cleavage of cyclic amines is known even for the electrochemical process since one could expect that the Shono-type oxidation products would readily undergo ring-cleavage under suitable conditions. Two important papers that the authors have not cited include one by Frankowski (*J. Org. Chem.* **2022**, *87*, 1173) and one by Sarpong (*J. Am. Chem. Soc.* **2023**, *145*, 11245). In the former work, Frankowski already showed that cyclic amines could be converted to acyclic amino aldehydes under electrochemical conditions and utilized this C–N bond cleavage approach for in-situ functionalization with isocyanide. In the latter work, Sarpong demonstrated that a similar type of transformations could be achieved using organo-photoredox catalysts and cheap oxidants. Given these precedents, the critical questions would be ‘what is new?’ and ‘what advantages does this method hold in comparison to the previous reports?’.

The compatibility of peptide motifs in this type of chemistry is not unusual either as demonstrated by White, who also showed similar post-modifications that involve the synthesis of macrocyclic peptides (ref. 47, see also the work by Higuchi; *J. Am. Chem. Soc.* **2005**, *127*, 834). The reaction scope, presented in this work, with respect to the peptide-containing substrates is rather simple and limited to piperidic rings, whereby the resulting amino aldehydes could, in fact, be afforded by other means as well, such as the peptide formation using 6-hydroxy-L-norleucine followed by oxidation (e.g. *Eur. J. Org. Chem.* **2018**, 6002). Considering these perspectives, the current work does not seem to offer synthetic advances, neither does it hold significance and novelty. Therefore, this reviewer does not recommend the publication of this work in Nature Communications.

Additional comments:

- There is nothing new in terms of mechanism, as one drawn in Fig. 7 is just based on the commonly proposed mechanism in this type of transformations. In this regard, this reviewer is not sure if the DFT calculations were necessary to understand the already well-established mechanism. Are there any new aspects in mechanistic perspectives?
- According to the computed k_{obs} , the substrates holding Piv-protecting group give the fastest reactions, but this does not really explain why other protecting groups (Cbz and Boc) are not effective at all. Looking at the numbers provided in Table 2, the reactions seem energetically feasible for all substrates, especially that the activation energies required for **Int7** to **TS4** are within similar ranges ($\Delta G^\ddagger = 12\text{--}14$ kcal/mol). Protonation has been suggested to play a

critical role in lowering the TS energy for the C–N bond cleavage, so could it simply be something to do with the basicity?

- For the investigation of potential racemization, the authors used chiral HPLC analysis and provided the corresponding HPLC traces in the Supplementary Information (pages 125–126), however, the data looks very odd. For **1b**, the HPLC traces provided for *rac-1b* does not show two signals in 1:1 ratio, indicating that the conditions used for the separation are inappropriate. In fact, *rac-1b* is not a mixture of racemates, but should rather be a mixture of diastereomers. How did the authors synthesize this mixture? Do they have NMR data for this mixture? Diastereomers should normally be distinguishable by NMR, so the authors should provide these details in the SI. For **65b**, once again, how did the authors synthesize *L-65b* and *rac-65b*? The HPLC traces for *L-65b* show two signals, which again seems inappropriate. The authors should re-optimize the conditions for the chiral HPLC, so that only single signal is obtained for each enantiomer. The HPLC traces should be provided with proper integration, in which the racemic mixtures should give two signals in 1:1 ratio.

Reviewer #2 (Remarks to the Author):

Ruan and Qiu's group depict an electrochemical ring-opening strategy for synthesis of unnatural peptide aldehydes via C–N bond cleavage of cyclic amines. The peptide aldehydes could be derivatized to generate some other novel peptide sequences. Furthermore, DFT calculations demonstrate the importance of N-Piv protecting group. Considering the peptide aldehydes may offer a valuable means for peptide backbone modification or site-specific ligation reactions, I would recommend publication following some revisions.

- 1: Considering the impact of employing different types of N-acyl groups on the reaction results, the cyclic voltammetry (CV) experiments should also investigate the oxidation peak potential of other different types of N-acyl groups (Boc, Bz, and Cbz).
- 2: The compatibility of this reaction should be improved. Please investigate some sensitive amino acids such as Ser, Met, Asn, Trp, Tyr, and Lys. If these amino acids are not compatible, please mention it and provide the explanation in the manuscript.
- 3: Please check Fig. 6c in the manuscript, oxidation peak potential of 1a, 1b, and 48 could not be found.
- 4: Please provide 1-2 examples for removing protective groups (Piv) from peptide products to compensate for the limitation of N-terminal derivatization.
- 5: Current should be expressed in mA/cm².

Reviewer #3 (Remarks to the Author):

The study conducted by Ruan and co-workers describe a mild electrochemically oxidative ring-opening of cyclic aliphatic amines to achieve peptide aldehydes, through selective C–N bond cleavage with H₂O as a nucleophile. The electro-synthetic protocol developed by the authors demonstrates a biocompatible, advanced, step- and atom-economical approach. Notably, it enables the generation of peptide aldehydes from (homo)proline-specific peptides as substrates through sustainable electrochemical oxidation, ensuring racemization-free conditions. For all I know, few studies have focused on the electrochemical modification of unnatural amino acids. This methodology holds significant promise in both academic research and industrial applications within medicinal discovery, offering an alternative

protocol that eliminates the need for protection and deprotection processes, as well as the use of metal-based chemical oxidants and harsh conditions.

To support their claims, the authors present a comprehensive substrate scope with lots of interesting examples. Furthermore, they emphasize the applicability of the method in late-stage modifications and the assembly of peptide-conjugated drugs (PDC), including both linear and macrocyclic peptides. The inclusion of mechanistic studies and density functional theory (DFT) calculations adds credibility to their findings, with a particular focus on the pivotal role played by the N-Piv protecting group.

Considering the overall significance and novelty of the work, I recommend its publication in Nature Communications. However, I suggest addressing the following comments to further enhance the clarity and completeness of the manuscript.

1. In the concluding paragraph of the introduction, the author asserts the formation of "high-yielding peptide aldehydes." However, upon a careful examination of the figures, it becomes evident that the yields achieved are more accurately described as moderate to good levels. Therefore, it is recommended to revise this statement in a manner that accurately reflects the observed outcomes.

2. Some latest references of electrochemical modification of tyrosine might be missing in the introduction. It is suggested to check the up-to date literature to ensure a comprehensive overview of the current state of research in this field.

3. In the section discussing control experiments, it is advisable to include testing the peptide substrates 1a and the peptide aldehyde product 1b. This addition will contribute to a more thorough examination of the methodology and enhance the credibility of the results.

Furthermore, it is noted that the text description of Figure 6c does not align accurately with the content of the figure. A revision is necessary to ensure that the text corresponds appropriately to the visual representation provided in Figure 6c.

Journal: *Nat. Commun.*

Title: Electrochemical Synthesis of Peptide Aldehydes via C–N Bond Cleavage of Cyclic Amines

Manuscript ID: NCOMMS-23-63936

Dear Editors and Reviewers,

Thank you for your letter and for the reviewers' comments concerning our manuscript entitled " Electrochemical Synthesis of Peptide Aldehydes via C–N Bond Cleavage of Cyclic Amines "(Manuscript ID: NCOMMS-23-63936), which we would like to resubmit and also wish to be considered for publication in Nature Communications. We have revised the manuscript and updated the SI according to reviewers and editor's suggestion, which satisfied the requirements asked by the editor. And the changes we have made also marked in yellow in the manuscript. Here are the responses point by point, which marked in blue.

Comments from the reviewers:

A. Responds to the reviewer 1

Recommendation: This reviewer does not recommend the publication of this work in Nature Communications.

Comments:

In this manuscript, Hu, Qiu, Ruan, and their co-workers report an electrochemical strategy for ring-opening of cyclic amines via oxidative C–N bond cleavage. The major reaction scope includes simple piperidine rings containing a range of peptide motifs, as well as amides and ester appendages, which leads to the corresponding acyclic amino aldehydes under mild reaction conditions. The reaction is also effective for cyclic amine structures of differed ring sizes, such as azetidines, pyrrolidines, azepines, and azocanes. The synthetic value of this development has been proven by further modification of the furnished amino aldehydes via reduction, fluorination, azidation, olefination, oxidation, esterification, and synthesis of macrocyclic peptides. Furthermore, the authors carried out the mechanistic investigation using labeling experiments and DFT calculations.

Question 1: Despite the extensive synthetic studies with a wide range of reaction scope and applications, the major concern arises on the originality of the presented chemistry. Oxidative C–N bond cleavage of cyclic amines is known even for the electrochemical process since one could expect that the Shono-type oxidation products would readily undergo ring-cleavage under suitable conditions. Two important papers that the authors have not cited include one by Frankowski (*J. Org. Chem.* 2022, 87, 1173) and one by Sarpong (*J. Am. Chem. Soc.* 2023, 145, 11245). In the former work, Frankowski

already showed that cyclic amines could be converted to acyclic amino aldehydes under electrochemical conditions and utilized this C–N bond cleavage approach for in-situ functionalization with isocyanide. In the latter work, Sarpong demonstrated that a similar type of transformations could be achieved using organo-photoredox catalysts and cheap oxidants. Given these precedents, the critical questions would be ‘what is new?’ and ‘what advantages does this method hold in comparison to the previous reports?’

Response: We appreciate the professional and detailed comments provided by the reviewer. The reviewer referenced Frankowski's work (*J. Org. Chem.* **2022**, 87, 1173), which demonstrated the conversion of cyclic amines to acyclic amino aldehydes under electrochemical conditions for in-situ functionalization with isocyanide. However, upon closer examination of the data presented in the article, it is evident that only when the *N*-substituent is Bz, a trace amount of aldehyde peak is detected by the ¹H NMR. Furthermore, the absence of a complete NMR spectra of the aldehyde product suggests potential instability of the aldehyde intermediate under the reaction conditions. Notably, no isolated aldehyde products were reported in the article, even with the utilization of Boc protective groups, as evidenced in scheme 2 of the original article (*J. Org. Chem.* **2022**, 87, 1173). This likely necessitates further in-situ transformation with isocyanide as a nucleophile. Our study reveals a nuanced relationship between Shono-type oxidation and the *N*-substituents of cyclic amines, wherein different *N*-substituents yield distinct products. As demonstrated in our manuscript and supporting information, only Piv and Bz result in acyclic amino aldehydes, while Cbz and Boc yield *ortho*-hydroxylation products when H₂O is employed as the nucleophile (see the below schemes). Achieving this selectivity in practice is non-trivial, as evidenced by earlier studies where *o*-alkoxylation products were obtained when alcohols were used as nucleophiles (ref. 59, *J. Am. Chem. Soc.* **1975**, 97, 4264), and more recently, *ortho*-carbonylation products were observed when water was utilized as a nucleophile (ref. 61, *Angew. Chem. Int. Ed.* **2018**, 57, 6686).

It appears the reviewer may have overlooked citation [71] (the present version [73]), which is pivotal to our work (*J. Am. Chem. Soc.* **2023**, 145, 11245). This paper is indeed cited in our manuscript (see ref. 73-74). In contrast to other methods involving photosensitizers, metal catalysts, and equivalent chemical oxidants, our electrochemical approach offers economic and environmentally friendly advantages by

circumventing these reagents. Furthermore, our study not only encompasses simple cyclic amines but also explores peptide substrates—a relatively underdeveloped area in electrochemical synthetic research, as highlighted in the referenced reviews (ref. 33 and 63, *J. Am. Chem. Soc.* **2022**, 144, 23; *Chin. J. Org. Chem.* 2024, 44, 903). Our substrate scope extends to polypeptides, enabling the synthesis of diverse carbon chain peptide aldehydes and macrocyclic peptides with broad applicability.

In summary, key aspects of our strategy include:

- a) Reliable output of final product amino aldehydes in our developed electro-system.
- b) Revelation of the pivotal role of *N*-protective groups in Shono-type oxidation ring opening, supported by relevant theoretical calculations.
- c) Demonstration of the importance of electrolytes, particularly acidic electrolytes, in the ring-opening process of oxidation, especially for polypeptide substrates, as corroborated by experimental and theoretical findings.
- d) Utilization of exceedingly mild and biocompatible conditions devoid of metal catalysts, chemical oxidants, and racemization.
- e) Unparalleled substrate scope encompassing various polypeptides and complex bioactive molecules.
- f) Paving the way for versatile syntheses of macrocyclic peptides.

We trust that these points might address the reviewer's queries and underscore the novelty and significance of our work.

Question 2: The compatibility of peptide motifs in this type of chemistry is not unusual either as demonstrated by White, who also showed similar post-modifications that involve the synthesis of macrocyclic peptides (ref. 47, see also the work by Higuchi; *J. Am. Chem. Soc.* 2005, 127, 834). The reaction scope, presented in this work, with respect to the peptide-containing substrates is rather simple and limited to pipercolic rings, whereby the resulting amino aldehydes could, in fact, be afforded by other means as well, such as the peptide formation using 6-hydroxy-L-norleucine followed by oxidation (e.g. *Eur. J. Org. Chem.* 2018, 6002).

Response: Thank you for your valuable comments. White and Higuchi have indeed demonstrated the conversion of peptide-containing cyclic amines to linear amino aldehydes, subsequently applying them in the modification of simple aldehyde peptides. However, it is noteworthy that the generation of these aldehyde peptides in their work relied on metal catalysts and a plethora of oxidants, which may not align with the current environmental standards of green synthetic chemistry. In spite of these challenges, it is commendable that White and Higuchi's research has paved the way for advancements in the synthesis of linear amino aldehydes from peptide-containing cyclic amines, showcasing their dedication to exploring innovative synthetic methodologies in peptide chemistry.

Furthermore, the work by Higuchi (J. Am. Chem. Soc. 2005, 127, 834) primarily focused on the synthesis of *N*-acyl amino acids through oxidative C-N bond cleavage with pyridine *N*-oxides catalyzed by ruthenium porphyrin. This indicates that the generation of amino aldehydes in their reaction system was not straightforward. Aldehydes are known for their reactivity and susceptibility to oxidation, necessitating milder conditions for their synthesis. In Higuchi's study, only the seven-membered ring in the substrate yielded aldehydes, albeit in modest yield (30%), underscoring the challenges associated with their stability and the limited selectivity observed with amino acid products.

Additionally, as mentioned by the reviewers, alternative methods for obtaining amino aldehydes exist (Eur. J. Org. Chem. 2018, 6002). However, this approach typically involves multiple steps, including removal of the protecting groups using palladium carbon and hydrogen, followed by condensation reactions, and finally oxidation of the resulting alcohol to obtain the aldehyde. This methodology resembles the traditional approach mentioned in the introduction of our manuscript, involving deprotection after peptide elongation to reveal the aldehyde functionality. Such multi-step processes not only have limited applications but also may not be as efficient or environmentally friendly as our proposed methodology.

Question 3: There is nothing new in terms of mechanism, as one drawn in Fig. 7 is just based on the commonly proposed mechanism in this type of transformations. In this regard, this reviewer is not sure if the DFT calculations were necessary to understand the already well-established mechanism. Are there any new aspects in mechanistic perspectives?

Response: Thanks for your comment. We believed that our DFT calculations have furnished an exhaustive energy profiles for the hypothesized reaction mechanism, elucidated the function of the electrolyte *n*-Bu₄NHSO₄, acting as a proton base within this process, disclosed the critical reaction step influencing the overall rate of decyclization, and revealed the lower reactivity of Boc-, Bz-, Cbz-substituted substrates.

Moreover, in the response of the 4th question, by examining the calculated energies, analyzing electron and steric influences, and conducting MBO analysis, we concluded that the overall reactivity may be influenced significantly by the steric repulsion between the Piv group and the piperidine ring, through the torsion of acyl group, raising the basicity of the Int6 intermediates.

Question 4: According to the computed kobs, the substrates holding Piv-protecting group give the fastest reactions, but this does not really explain why other protecting groups (Cbz and Boc) are not effective at all. Looking at the numbers provided in Table 2, the reactions seem energetically feasible for all substrates, especially that the activation energies required for Int7 to TS4 are within similar ranges ($\Delta G^\ddagger = 12\text{--}14$ kcal/mol). Protonation has been suggested to play a critical role in lowering the TS

energy for the C–N bond cleavage, so could it simply be something to do with the basicity?

Response: Thanks for your comment. As we defined and explained in the Supplementary Information, the computed k_{obs} , which predicts the reaction rate constant of the overall reaction of iminium (**Int3**) decyclization: **Int3** + H₂O → Product, was related to the relative free energy difference between **TS4** and **Int3** + H₂O, which can be decomposed into $G_{\text{rel}}(\text{TS4}) - G_{\text{rel}}(\text{Int7})$ and $G_{\text{rel}}(\text{Int7}) - G_{\text{rel}}(\text{Int3})$. Although the activation energies from **Int7** to **TS4** ($G_{\text{rel}}(\text{TS4}) - G_{\text{rel}}(\text{Int7})$) of different substrates are close (12~13 kcal/mol, Table 2), there lie significant differences in the relative free energy of the **Int7** intermediate ($G_{\text{rel}}(\text{Int7}) - G_{\text{rel}}(\text{Int3})$, 5~11 kcal/mol, Table 1), strongly contributing to the differences in the k_{obs} .

Because the G_{rel} of **Int7** and the protonation energy changes ($G_{\text{rel}}(\text{Int7}) - G_{\text{rel}}(\text{Int6})$, -0.5~5 kcal/mol, Table 2) differ prominently, we assumed that the G_{rel} of **Int7** was greatly influenced by the basicity of N atom in **Int6**, determined by electronic effect and steric effect.

To consider the electronic effect, we calculated the energies of acetyl-substituted intermediates (Ac, Table 1), where the electronic effect of Ac group is similar with the Piv group. According to results, $G_{\text{rel}}(\text{Int7-Ac})$ and $G_{\text{rel}}(\text{TS4-Ac})$ were prominently higher than $G_{\text{rel}}(\text{Int7-Piv})$ and $G_{\text{rel}}(\text{TS4-Piv})$. Therefore, when the electronic effect of Acyl groups are similar, the steric effect may make a great impact. Furthermore, $G_{\text{rel}}(\text{Int7-Ac})$ was observed lower than $G_{\text{rel}}(\text{Int7-Cbz})$ and $G_{\text{rel}}(\text{Int7-Boc})$, indicating that a less electron-withdrawing group might raise the $G_{\text{rel}}(\text{Int7})$.

The steric effect may arise from the repulsion force between the acyl group and the piperidine ring, which may contribute to the torsion of the acyl group in **Int6** intermediates. The torsion of acyl renders the nitrogen atom in **Int6** donating less electron density to the carbonyl through conjugation, thus leading to the improvement of the basicity of the N atom. In Figure R1, we defined a dihedral $d(\text{C1-N-C2-O})$ for **Int6** to measure the extent of such torsion. Shown in Table 3, when the *N*-acyl is formyl (CHO), the steric hindrance is too negligible to render a torsion, and thus $d(\text{C1-N-C2-O})$ is close to 180°. As for **Int6-Bz** and **Int6-Piv**, $d(\text{C1-N-C2-O})$ is around 140°, indicating an obvious acyl torsion, preventing the conjugation between the nitrogen atom and the carbonyl.

To characterize the extent of the conjugation between N atom and the carbonyl, the calculated Mayer bond order (MBO) of C–N bond may be an appropriate descriptor. Obviously, the MBO of C–N bond in **Int6-Piv** intermediate is lower than others significantly (1.211, Table 3), indicating lower conjugation electron donation and stronger basicity.

In addition, the corresponding *ortho*-hydroxylation products can be obtained by

converting the *N*-acyl group into Cbz or Boc (Figure 2 and 3). Further attempts to change the electrolyte to basic *n*-Bu₄NOH under standard conditions resulted in reduced yields of both products 1b (29%) and 63b (72%).

To summarize, through analyses on calculated energies, interpretation of electron and steric effects, and MBO analysis, we believed that the steric hindrance between Piv group and the piperidine ring, and the ensuing torsion of acyl group, is the main factor in increasing the basicity of the nitrogen atom in **Int6**, and hence lower the $G_{\text{rel}}(\mathbf{Int7-Piv})$, which can provide a better explanation on the better reactivity of the Piv-substituted reactants.

Figure 1. Left: the definition of dihedral $d(\text{C1-N-C2-O})$ in **Int6**, labelled in red dashed lines; Medium: the defined C–N bond to analyze Mayer bond order (MBO), in red dashed lines; Right: The 3D-structure of the **Int6-Piv** intermediate.

Table 1. Relative Gibbs free energies (G_{rel}) of selected intermediates, with different *N*-acyl groups. Energy unit: kcal/mol.

N -acyl Group & G_{rel}	Int3	TS2	Int5	Int6	Int7	TS4
Ac	0.0	15.19	8.11	5.25	8.42	21.58
Boc	0.0	16.10	13.24	6.58	10.77	23.63
Bz	0.0	14.31	9.47	5.10	8.08	20.67
Cbz	0.0	15.15	14.85	4.74	9.69	22.34
Piv	0.0	14.70	8.98	5.92	5.66	17.78

Table 2. Gibbs free energies differences between selected intermediates, with different *N*-acyl groups. Energy unit: kcal/mol.

N -acyl Group	$G_{\text{rel}}(\mathbf{TS3}) - G_{\text{rel}}(\mathbf{Int7})$	$G_{\text{rel}}(\mathbf{Int7}) - G_{\text{rel}}(\mathbf{Int6})$
Ac	13.16	3.17
Boc	12.86	4.19
Bz	12.55	2.98
Cbz	12.65	4.95
Piv	12.12	-0.26

Table 3. Dihedral $d(\text{C1-N-C2-O})$ and Mayer bond order (MBO) of C–N bond, of different **Int6** intermediates.

N -acyl Group	Dihedral $d(\text{C1-N-C2-O})$ (°)	MBO of C–N bond
Ac	153.9	1.262

Boc	155.5	1.283
Bz	138.5	1.223
Cbz	169.9	1.276
CHO	179.3	1.278
Piv	142.1	1.211

Benzyl 2-hydroxypiperidine-1-carboxylate: $^1\text{H NMR}$ (500 MHz, $\text{DMSO-}d_6$) δ = 7.41 – 7.28 (m, 5H), 5.63 (s, 2H), 5.11 – 5.00 (m, 2H), 3.74 (d, J = 12.2 Hz, 1H), 3.05 (t, J = 12.7 Hz, 1H), 1.77 – 1.57 (m, 3H), 1.77 – 1.57 (m, 2H), 1.37 – 1.28 (m, 1H). $^{13}\text{C NMR}$ (125 MHz, $\text{DMSO-}d_6$) δ = 154.5, 137.0, 128.5, 127.9, 127.6, 73.6, 66.1, 38.5, 31.2, 24.8, 17.6. HR-MS(ESI) m/z calcd for $\text{C}_{13}\text{H}_{17}\text{NO}_3\text{Na}^+$ $[\text{M}+\text{Na}]^+$ 258.1101, found 258.1099.

Figure 2. The NMR spectrum and HR-MS(ESI) spectrum for *Int6-Cbz*.

Tert-butyl 2-hydroxypiperidine-1-carboxylate: ^1H NMR (400 MHz, DMSO- d_6) δ = 5.53 (s, 1H), 5.44 (d, J = 3.2 Hz, 1H), 3.66 (d, J = 12.7 Hz, 1H), 2.94 (t, J = 12.0 Hz, 1H), 1.79 – 1.52 (m, 4H), 1.48 – 1.42 (m, 2H), 1.40 (s, 9H). ^{13}C NMR (100 MHz, DMSO- d_6) δ = 153.5, 78.4, 72.9, 37.9, 31.1, 27.9, 24.7, 17.4. HR-MS(ESI) m/z calcd for $C_{10}H_{19}NO_3Na^+$ $[M+Na]^+$ 224.1257, found 224.1254.

Figure 3. The NMR spectrum and HR-MS(ESI) m/z spectrum for *Int6-Boc*.

Question 5a: For the investigation of potential racemization, the authors used chiral HPLC analysis and provided the corresponding HPLC traces in the Supplementary Information (pages 125–126), however, the data looks very odd. For **1b**, the HPLC traces provided for rac-**1b** does not show two signals in 1:1 ratio, indicating that the conditions used for the separation are inappropriate. In fact, rac-**1b** is not a mixture of racemates, but should rather be a mixture of diastereomers. How did the authors synthesize this mixture? Do they have NMR data for this mixture? Diastereomers should normally be distinguishable by NMR, so the authors should provide these details in the SI.

Response: Thank you for the valuable comments and suggestions. We acknowledge the validity of the reviewer's observation regarding the HPLC traces provided for rac-**1b**, and we agree that it may not be suitable for judging racemization. We apologize for not finding a suitable HPLC separation method. Due to the weak fluorescence of the compound, detection via evaporative photodetector was necessary. Consequently, we have relied on nuclear magnetic resonance (NMR) spectroscopy for our analysis, as suggested by the reviewers. Our NMR data indicates that the NMR spectra of **1b** with a single configuration appear singular, with a diastereomeric ratio (*dr*) value exceeding 20:1, whereas diastereomers spectra appear mixed, with a *dr* value of approximately 3.6:1. Based on these findings, we confidently conclude that our electrochemical oxidation method does not induce racemization. We have revised and updated in the supporting information.

Methyl [(R)-6-oxo-2-pivalamidohexanoyl]-L-leucinate (1b): ^1H NMR (400 MHz, CDCl_3) $\delta = 9.75$ (s, 1H), 7.01 (d, $J = 8.1$ Hz, 1H), 6.42 (d, $J = 7.7$ Hz, 1H), 4.59 – 4.51 (m, 1H), 4.51 – 4.43 (m, 1H), 3.70 (s, 3H), 2.57 – 2.46 (m, 2H), 1.93 – 1.82 (m, 1H), 1.73 – 1.52 (m, 6H), 1.21 (s, 9H), 0.93 (d, $J = 4.9$ Hz, 3H), 0.91 (d, $J = 4.9$ Hz, 3H). ^{13}C NMR (100 MHz, CDCl_3) $\delta = 202.2, 179.1, 173.2, 171.6, 52.42, 52.40, 50.9, 43.4, 41.3, 38.9, 31.5, 27.6, 25.0, 22.9, 21.8, 17.7$.

Methyl (6-oxo-2-pivalamidohexanoyl)-L-leucinate (dia-1b, $dr = 3.6:1$): ^1H NMR (400 MHz, CDCl_3) $\delta = 9.72$ (s, 1H), 7.30 (d, $J = 8.1$ Hz, 1H), 6.52 (d, $J = 7.7$ Hz, 1H), 4.59 – 4.43 (m, 2H), 3.67 (s, 3H), 2.56 – 2.41 (m, 2H), 1.92 – 1.75 (m, 1H), 1.74 – 1.48 (m, 6H), 1.19 (s, 9H), 0.91 (d, $J = 6.2$ Hz, 3H), 0.88 (d, $J = 6.2$ Hz, 3H). ^{13}C NMR (100 MHz, CDCl_3) $\delta = 202.3, 179.0, 173.2, 171.7, 52.4, 52.3, 50.7, 43.4, 41.1, 38.8, 31.8, 27.5, 24.9, 22.9, 21.6, 17.6$.

Question 6b: For 65b, once again, how did the authors synthesize L-65b and rac-65b? The HPLC traces for L-65b show two signals, which again seems inappropriate. The authors should re-optimize the conditions for the chiral HPLC, so that only single signal is obtained for each enantiomer. The HPLC traces should be provided with proper integration, in which the racemic mixtures should give two signals in 1:1 ratio.

Response: Thank you for your valuable suggestions. We have once again synthesized compound *rac-66b* (*rac-65b* in previous version) from piperidine-2-carboxylic acid for conducting C–N bond activation of cyclic amines. Subsequently, we reanalyzed the compound using chiral HPLC, employing a revised analysis method. Upon reevaluation, we identified errors in the previous analysis method and have rectified these in the Supplementary Information. Through HPLC analysis, it can also be found that our method does not occur racemization. We appreciate your attention to detail and apologize for any confusion caused.

HPLC chromatograms were recorded on an UltiMate3000 Infinity using the column CHIRALPAK® ID and H₂O/MeOH (50:50, 0.5 mL/min, detection at 199nm/UV).

< Chromatogram >

< Column Performance Report >

Peak No.	Time	Area	Area %	Plate number	Tailing	Resolution
1	17.441	8790134	49.874	2184	1.101	--
2	20.701	8834701	50.126	965	1.301	1.568

< Chromatogram >

< Column Performance Report >

Peak No.	Time	Area	Area %	Plate number	Tailing	Resolution
1	17.197	16346064	98.066	2031	1.147	--
2	20.567	322321	1.934	1243	1.208	1.747

The overlap HPLC chromatograms of *rac*-66b and 66b

B. Respond to the 2st reviewer

Recommendation: Considering the peptide aldehydes may offer a valuable means for peptide backbone modification or site-specific ligation reactions, I would recommend publication following some revisions.

Comments:

Ruan and Qiu's group depict an electrochemical ring-opening strategy for synthesis of unnatural peptide aldehydes via C–N bond cleavage of cyclic amines. The peptide aldehydes could be derivatized to generate some other novel peptide sequences. Furthermore, DFT calculations demonstrate the importance of N-Piv protecting group.

Question 1: Considering the impact of employing different types of N-acyl groups on the reaction results, the cyclic voltammetry (CV) experiments should also investigate the oxidation peak potential of other different types of N-acyl groups (Boc, Bz, and Cbz).

Response: Thanks for your suggestions. As shown in the figure above, the CV experiment was investigated to test the potential oxidation peaks of Boc, Cbz, and Piv, and found that the *N*-Boc oxidation potential was the lowest. This result was consistent

with our calculation. We have revised and updated in the supporting information.

(i) Red line: **63a** (1 mM); (j) Purple line: *tert*-butyl piperidine-1-carboxylate (1 mM); (k) Blue line: **62a** (1 mM); (l) Green line: benzyl piperidine-1-carboxylate (1 mM); (m) Black line: background. 0.1 M $n\text{Bu}_4\text{NPF}_6$ was dissolved in MeCN.

The cyclic voltammetry (CV) experiments of several types of *N*-acyl groups (Boc, Bz, and Cbz)

Question 2: The compatibility of this reaction should be improved. Please investigate some sensitive amino acids such as Ser, Met, Asn, Trp, Tyr, and Lys. If these amino acids are not compatible, please mention it and provide the explanation in the manuscript.

Response: We attempted reactions with these substrates and observed negative results for the unprotected amino acids. However, we were able to obtain the product for the protected Lysine and Serine substrates. We have revised in the manuscript and SI.

NP: no desired product SM: starting material

Question 3: Please check Fig. 6c in the manuscript, oxidation peak potential of 1a, 1b, and 48 could not be found.

Response: We have supplemented the cyclic voltammograms of the relevant compounds in the manuscript and SI, please check it. Thanks.

Question 4: Please provide 1-2 examples for removing protective groups (Piv) from peptide products to compensate for the limitation of N-terminal derivatization.

Response: Thanks for your suggestions. We successfully removed the Piv protective group from product 63b, yielding product 77. We have revised and updated in the manuscript and supporting information.

5-Aminopentanoic acid (77): ^1H NMR (400 MHz, $\text{DMSO-}d_6$) δ = 8.15 (s, 3H), 2.73 (t, J = 6.2 Hz, 2H), 2.23 (t, J = 6.6 Hz, 2H), 1.66-1.44 (m, 2H). ^{13}C NMR (100 MHz, $\text{DMSO-}d_6$) δ = 174.1, 38.4, 33.2, 26.4, 21.5. HR-MS(ESI) m/z calcd for $\text{C}_5\text{H}_{12}\text{NO}_2$ $[\text{M}+\text{H}]^+$ 118.0863, found 118.0863.

Question 5: Current should be expressed in mA/cm^2 .

Response: We have revised in the manuscript and SI. Thanks.

C. Respond to the 3st reviewer

Recommendation: Considering the overall significance and novelty of the work, I recommend its publication in Nature Communications. However, I suggest addressing the following comments to further enhance the clarity and completeness of the manuscript.

Comments:

The study conducted by Ruan and co-workers describe a mild electrochemically oxidative ring-opening of cyclic aliphatic amines to achieve peptide aldehydes, through selective C–N bond cleavage with H_2O as a nucleophile. The electro-synthetic protocol developed by the authors demonstrates a biocompatible, advanced, step- and atom-economical approach. Notably, it enables the generation of peptide aldehydes from (homo)proline-specific peptides as substrates through sustainable electrochemical oxidation, ensuring racemization-free conditions. For all I know, few studies have focused on the electrochemical modification of unnatural amino acids. This methodology holds significant promise in both academic research and industrial applications within medicinal discovery, offering an alternative protocol that eliminates the need for protection and deprotection processes, as well as the use of metal-based chemical oxidants and harsh conditions. To support their claims, the authors present a

comprehensive substrate scope with lots of interesting examples. Furthermore, they emphasize the applicability of the method in late-stage modifications and the assembly of peptide-conjugated drugs (PDC), including both linear and macrocyclic peptides. The inclusion of mechanistic studies and density functional theory (DFT) calculations adds credibility to their findings, with a particular focus on the pivotal role played by the N-Piv protecting group.

Question 1: In the concluding paragraph of the introduction, the author asserts the formation of "high-yielding peptide aldehydes." However, upon a careful examination of the figures, it becomes evident that the yields achieved are more accurately described as moderate to good levels. Therefore, it is recommended to revise this statement in a manner that accurately reflects the observed outcomes.

Response: Thanks a lot for your comments. We have revised in the manuscript. Please check it.

Question 2: Some latest references of electrochemical modification of tyrosine might be missing in the introduction. It is suggested to check the up-to date literature to ensure a comprehensive overview of the current state of research in this field.

Response: Thanks for your suggestions. The literatures of electrochemical modification of tyrosine have been cited as ref. [40-45], respectively. (Chem. Sci., 2021, 12, 15374–15381; Org. Chem. Front., 2023, 10, 4606–4615)

Question 3: In the section discussing control experiments, it is advisable to include testing the peptide substrates 1a and the peptide aldehyde product 1b. This addition will contribute to a more thorough examination of the methodology and enhance the credibility of the results. Furthermore, it is noted that the text description of Figure 6c does not align accurately with the content of the figure. A revision is necessary to ensure that the text corresponds appropriately to the visual representation provided in Figure 6c.

Response: Thanks for your suggestions. We have revised in the manuscript and SI. Please check it.

Isotopic Labeling Experiment

REVIEWERS' COMMENTS

Reviewer #1 (Remarks to the Author):

While the authors have addressed most of the concerns raised by this reviewer in the previous round of peer review, this reviewer is still skeptical about the DFT calculation part. The computational outcomes indeed suggest that the reaction would be energetically more favourable with the Piv group, but other N-protecting groups also give reasonable energies and the related ring-opening processes seem still possible. For example, going from **int6** to **int7** and from **int7** to **TS2**, N-Boc substrate shows similar energy profile compared to N-Bz substrate with very marginal energy differences, but contrasting results were obtained experimentally (the former only gives a hydroxylation product whereas the latter furnishes a ring-opening aldehyde product). The basicity-related rationale does make much sense, but the calculated energies do not in some parts. Yet, this could just be this reviewer being too picky, hence, this reviewer would still recommend the publication of this manuscript in Nature Communications if other reviewers have no issues with the DFT calculations.

Reviewer #2 (Remarks to the Author):

In their revised version, the authors answered my concerns satisfyingly and the changes done are sufficient for me. Considering those sensitive amino acids such as Ser, Met, Asn, Trp, Tyr, and Lys are not compatible, I would suggest to remove or change the following sentence: "unparalleled broad substrate scope with various polypeptides and complex bioactive molecules".

Reviewer #3 (Remarks to the Author):

All the concerns of this referee have satisfactorily been addressed. Hence, I recommend accepting the manuscript in its present form.

The point-by-point response to the reviewers' comments

Reviewer #1 (Remarks to the Author):

While the authors have addressed most of the concerns raised by this reviewer in the previous round of peer review, this reviewer is still skeptical about the DFT calculation part. The computational outcomes indeed suggest that the reaction would be energetically more favourable with the Piv group, but other N-protecting groups also give reasonable energies and the related ring-opening processes seem still possible. For example, going from **int6** to **int7** and from **int7** to **TS2**, N-Boc substrate shows similar energy profile compared to N-Bz substrate with very marginal energy differences, but contrasting results were obtained experimentally (the former only gives a hydroxylation product whereas the latter furnishes a ring-opening aldehyde product). The basicity-related rationale does make much sense, but the calculated energies do not in some parts. Yet, this could just be this reviewer being too picky, hence, this reviewer would still recommend the publication of this manuscript in Nature Communications if other reviewers have no issues with the DFT calculations.

Response: Shown in **Table R1**, for variant *N*-substituents, the free energy difference between TS4 and Int6 can be considered as the ring-opening energy barrier, deciding the rate of C–N cleavage. Regarding the *N*-Boc and *N*-Bz intermediates, $G(\text{TS4}) - G(\text{Int6})$ of the former is 1.48 kcal/mol higher, indicating that the C–N cleavage rate of *N*-Bz intermediates is around 12 times ($e^{1.48 \times 4.184 \div 8.314 \div 298.15 \times 1000} \approx 12$) greater than that of *N*-Boc intermediates. The significant difference in C–N cleavage rate may support experimental results where *N*-Bz substrates were converted into aldehydes, while *N*-Boc substrates underwent hydroxylation.

Moreover, the calculated k_{obs} for *N*-Boc intermediates ($7.2 \times 10^{-7} \text{ L}\cdot\text{mol}^{-1}\cdot\text{s}^{-1}$) indicates a slow overall rate of iminium's ring-opening, while the calculated k_{obs} for *N*-Bz intermediates ($8.8 \times 10^{-4} \text{ L}\cdot\text{mol}^{-1}\cdot\text{s}^{-1}$) is prominently higher, which may be a proper explanation to the differences of the products.

Our proposed mechanism and DFT calculations can well-disclosure the backbone

changes of intermediates along the reaction pathway. Therefore, among various *N*-substituted intermediates, the calculated differences in $G(\text{TS4}) - G(\text{Int6})$ and the k_{obs} should make sense in revealing the reactivity.

Table R1. Relative Gibbs free energies (G_{rel}) of selected intermediates (or TSs), and the value of $G(\text{TS4}) - G(\text{Int6})$, with different *N*-acyl groups. Energy unit: kcal/mol.

N -acyl Group & G_{rel}	Int6	Int7	TS4	$G(\text{TS4}) - G(\text{Int6})$
Boc	6.58	10.77	23.63	17.05
Bz	5.10	8.08	20.67	15.57
Cbz	4.74	9.69	22.34	17.60
Piv	5.92	5.66	17.78	11.86

Reviewer #2 (Remarks to the Author):

In their revised version, the authors answered my concerns satisfyingly and the changes done are sufficient for me. Considering those sensitive amino acids such as Ser, Met, Asn, Trp, Tyr, and Lys are not compatible, I would suggest to remove or change the following sentence: “unparalleled broad substrate scope with various polypeptides and complex bioactive molecules” .

Response: Thanks a lot for your comments. We have deleted “unparalleled”, and revised to “ii) a broad substrate scope with various peptides and complex bioactive molecules,” in the manuscript. Please check it.

Reviewer #3 (Remarks to the Author):

All the concerns of this referee have satisfactorily been addressed. Hence, I recommend accepting the manuscript in its present form.

Response: We highly appreciate the reviewer’s positive comments on our manuscript. We are sure that the quality of this work has been greatly improved according to these nice comments and wise suggestions. Thanks very much.